# Positive Impact, Creativity, and Innovative Behavior at Work: The Mediating Role of Basic Needs Satisfaction

**DOI:** 10.3390/bs13120984

**Published:** 2023-11-28

**Authors:** Konstantinos Papachristopoulos, Marc-Antoine Gradito Dubord, Florence Jauvin, Jacques Forest, Patrick Coulombe

**Affiliations:** 1Department of Theoretical Studies, Athens School of Fine Arts, 10682 Athens, Greece; 2Future of Work Institute, Curtin University, Perth, WA 6000, Australia; marc-antoine.gradito-dubord@uqtr.ca; 3Department of Human ReSources Management, Université du Québec à Trois-Rivières, Trois-Rivières, QC G8Z 4M3, Canada; 4École des Sciences de la Gestion, Université du Québec à Montréal, Montréal, QC H2X 3X2, Canada; jauvin.florence@courrier.uqam.ca (F.J.); forest.jacques@uqam.ca (J.F.); coulombe.patrick@uqam.ca (P.C.)

**Keywords:** positive impact, needs satisfaction, prosocial motivation, benevolence, innovative work behavior, creativity

## Abstract

In recent research, a growing body of empirical evidence suggests that prosocial impact at work can play a significant role in enhancing creativity and innovativeness. Drawing from self-determination theory, we hypothesized that basic psychological needs and benevolence satisfaction could serve as a mediating factor in the relation between an employee’s perceived social impact and innovative work behavior and creativity, thus illuminating the manner in which the contentment of psychological needs fosters inventive proclivities within the organizational milieu. Results from a study in Greece and Canada (N = 528) showed that both perceived social impact and prosocial motivation are positively associated with innovative work behavior and creativity while autonomy and competence satisfaction mediate the relation between perceived social impact and the work outcomes examined within this study. Moreover, prosocial motivation was found to moderate the relation between benevolence satisfaction and innovativeness. Findings extend prior research on the role of prosociality on creative behavior at work and provide supporting evidence for the organizations that encourage and support employees’ initiatives to make a positive difference in the lives of others.

## 1. Introduction

Amidst the contemporary, fast-paced, and dynamic landscape of the modern workplace, delving into the investigation of innovative work behavior (IWB) could potentially provide valuable insights for achieving organizational success [1]. IWB, as defined by Janssen [2], involves the intentional creation, introduction, and application of new ideas within a work role, group, or organization, aimed at benefiting role performance, the group, or the organization. This multifaceted concept encompasses activities such as problem and/or solution searching, idea generation, idea championing (i.e., attempting to build support for ideas), and idea implementation [3].

At the core of innovative work behaviors (IWB) resides the fundamental essence of creativity. Creativity, as defined by Amabile [4] and Van Dyne et al. [5], emphasizes the generation of original and novel work, focusing on the creation of new and innovative ideas. However, the concept of IWB encompasses a broader scope. IWB necessitates more than mere displays of creative behavior. Since IWB refers mostly to a set of behavioral tasks that help employees develop, promote, and implement new and innovative ideas, [6,7] broaden the concept beyond creativity. [1] IWB includes actions that are not always normally rewarded and rewarding for an employee since championing ideas and supporting new services/products involves energy, personal investment, and a sense of personal agency, and to date, relatively few studies have focused on the individual-level mechanisms of IWB [8]. Innovation involves trials and failures, feelings of having autonomy to act, and energy to try out new ideas repeatedly with less fear of being judged, and we can assume that these energetic resources need to be aligned with high-level and high-quality motivational states [1,9]. Scott and Bruce [10] pointed out that motivation lies at the heart of innovative work behavior, and therefore, understanding the motivational antecedents of IWB is crucial [11,12,13]. This dynamic interplay of these multifaceted elements suggests that energetic resources must align with high-level and high-quality motivational states to effectively foster and sustain IWB.

In recent research, a growing body of empirical evidence, primarily focusing on management applications, suggests that prosocial motivation can play a significant role in enhancing creativity and innovativeness [14]. Prosocial motivation pertains to the desire to act for the benefit and welfare of others and has been linked to various positive personal and professional outcomes [15,16]. Zhang and Bartol [17] proposed that the extent of an individual’s engagement in the innovative process depends on their concern for the problem at hand. Consequently, employees with prosocial motivation, driven by the greater interests of others, the organization, or groups, exhibit a heightened focus on the well-being and needs of others [18]. Investigating the role of prosocial motivation, Grant and Berry [19] underscore the significance of directing employees’ attention towards the development and evaluation of ideas that effectively benefit beneficiaries. This highlights the notion of prosocial impact.

Prosocial impact is characterized by individuals’ perceptions of their work behavior benefiting others, a perception largely influenced by the meaningfulness of their work content [1]. It helps employees go beyond the limitations of their own perspectives, improve their sensitivity to the needs of others, and perform tasks to the best of their abilities and interests [20], all of which are crucial for promoting innovativeness and creativity [21]. Zhang and Bartol [17] suggested that the degree of individual participation in the innovative process depends on the degree of his/her concern regarding the problem, and thus, prosocially motivated employees who are driven by the greater interests of others, the organization, or groups are more concerned about the well-being and needs of others [18]. Also, in a series of four experiments, Polman and Emich [22] demonstrated that creative performance was significantly higher when participants were instructed to make creative decisions on the behalf of others rather than for the self. In another series of three studies, [22] also conducted experiments with three-person groups that performed brainstorming tasks for solving specific problems under different motivational conditions. Similar to Grant and Berry’s [19] results, Bechtholdt, De Dreu, Nijstad, and Choi [23] found that groups who expected an evaluation of overall group performance (prosocial motive scenario) showed higher ideational originality than groups that expected incentives for each member’s contribution (pro-self-motive condition).

Prosocial impact allows employees to transcend their own perspectives, heightens their sensitivity to the needs of others, and enables them to perform tasks with the utmost dedication and interest, all of which are crucial elements in fostering creativity and innovativeness [24]. However, since not all employees may possess inherent self-regulation or prosocial motivation, organizations have a responsibility to provide opportunities for employees to experience meaningful work [25]. Furthermore, existing research suggests that perceiving one’s work as positively impacting others’ well-being serves as a significant need-satisfying factor across diverse cultural and occupational contexts [26]. The current study aims at addressing some recent calls asserting that prosocial motivation is a potentially important yet understudied determinant of innovative behavior deserving of further applied research [27,28].

The primary goal of this study is to utilize basic psychological needs [29] as a framework to explore the underlying mechanisms that could explain the potential relationship between prosocial impact, prosocial motivation, innovative work behaviors, and creativity. By employing this theoretical lens, the study aims to shed light on the intricate processes through which prosocial impact and motivation may influence employees’ innovative work behaviors and creative outputs.

The notion of basic psychological needs satisfaction (BPNS), derived from self-determination theory [30], emerges as a potential mechanism to elucidate the underlying dynamics that govern the relationship between prosocial impact, innovative work behaviors (IWB), and creativity. The three fundamental psychological needs are as follows: autonomy, competence, and relatedness. Autonomy represents the desire for individuals to experience a sense of independence and choice in their actions, fostering a feeling of control over their decisions aligned with personal values and interests. Competence reflects the need for individuals to feel effective and skilled in their pursuits, contributing to a sense of mastery and accomplishment. Lastly, relatedness pertains to the need for individuals to experience social connectedness, care, and support in their relationships, fostering a sense of belongingness and understanding. These core psychological needs are pivotal in promoting intrinsic motivation, well-being, and optimal functioning across various domains of life. Indeed, scholarly discourse has posited that a positive impact may be associated with an increase in the fulfillment of basic psychological needs [31]. Substantiating this supposition, compelling empirical findings underscore the intricate connection between basic psychological need satisfaction (BPNS), innovative work behaviors (IWB), and creativity, thus illuminating the manner in which contentment of psychological needs fosters inventive proclivities and creativity within the organizational milieu [32,33].

**Hypothesis 1 (H1).** *Through greater need satisfaction, prosocial impact leads to greater innovative work behaviors (IWB) and creativity.*

According to Martela and Ryan [20], benevolence—the act of positively contributing to others—could potentially be considered as a “fourth” psychological need. Controlling for the three initial needs (i.e., autonomy, competence, and relatedness), they demonstrated through three independent studies that benevolence satisfaction acts an indirect effect in the relationship between prosocial impact and well-being, with all four factors emerging as independent constructs [34]. Subsequent studies have further shown that satisfaction of benevolence significantly assists individuals in finding meaning at work [35]. Recent research suggests that instead of being considered a fundamental psychological need, benevolence may be viewed as a well-being enhancer [36]. This shift in perspective arises from the unclear construct validity of benevolence frustration. [37]. Well-being enhancers are characterized as “universal conditions for enhancing human flourishing, wherein satisfaction should lead to optimal development and overall well-being” [36]. However, their frustration might not necessarily have distinct effects on causing ill-being. Based on the preceding findings, exploring the indirect role of benevolence satisfaction in the relationship between prosocial impact, innovative work behaviors (IWB), and creativity holds significant promise for understanding essential organizational outcomes.

**Hypothesis 2 (H2).** *Through greater benevolence satisfaction, prosocial impact leads to greater innovative work behaviors (IWB) and creativity.*

Moreover, considering that not all employees are inherently self-regulated or prosocially motivated [25], it becomes imperative to acknowledge the potential role of prosocial motivation in amplifying the relationships between benevolence satisfaction, innovative work behaviors (IWB), and creativity. IWB includes actions that are not always normally rewarded and rewarding for an employee, since championing ideas and supporting new services/products involves energy, personal investment, and a sense of personal agency. Prosocial motivation refers to the desire to act for the benefit or welfare of others and has been linked to a wide array of positive personal and professional outcomes [15] since it helps employees go beyond the limitations of their own perspectives, improve their sensitivity to the needs of others, and perform tasks to the best of their abilities and interests, all of which are crucial for promoting innovativeness and creativity [24]. The degree of individual participation in the innovative process depends on the degree of his/her concern regarding the problem and thus, prosocially motivated employees who are driven by the greater interests of others, the organization, or groups are more concerned about the well-being and needs of others [17,38].

**Hypothesis 3 (H3).** *The relation between benevolence satisfaction, innovative work behaviors, and creativity is moderated by prosocial motivation.*

This study aims to make valuable contributions to the field of human resources and organizational behavior. Firstly, it builds an integrated model, drawing from the self-determination theory (SDT) and prior research, to investigate the connections between prosocial impact, needs satisfaction, prosocial motivation, and innovative work behaviors (IWB) and creativity. Secondly, it aims to highlight if prosocial motivation goes beyond idea generation; the research examines whether needs satisfaction and benevolence satisfaction have an indirect effect in the link between prosocial impact and IWB and creativity. Additionally, the study explores the moderating role of prosocial motivation in the relationship between benevolence satisfaction and IWB. By adopting this comprehensive approach, the study seeks to provide valuable insights into the complex dynamics of these variables and their implications for human resources.

## 2. Materials and Methods

### 2.1. Participants and Procedures

In this study, we recruited 528 employees from various industries and occupations in Greece and Canada. One part of the Canadian sample (N = 309) was recruited using a convenience sampling method (N = 118) while the rest of the participants were recruited on Prolific Academic (N = 191) with criteria to ensure their resemblances to our population (i.e., French-speaking, living in Canada). The Greek sample (N = 219) was recruited using a convenience sampling method. To ensure sufficient statistical power, we calculated the sample size using G*Power v3.1 software with an effect size of 0.15 and a power of 0.95.

### 2.2. Measures

#### 2.2.1. Innovative Work Behavior

Innovative work behavior was measured by nine items adapted from De Jong and Den Hartog’s [3]. The IWB scale is a unidimensional measure that incorporates items to reflect four stages of IWB, i.e., exploration, generation, championing, and implementation of ideas. Participants were required to indicate how frequently, using a 7-point Likert-type scale ranging from 1 (almost never) to 7 (almost always), they manifest the behaviors mentioned in the survey. A sample item is “how often do you find new approaches to execute tasks?”. Cronbach’s alpha coefficient was 0.90.

#### 2.2.2. Creativity

Creativity was assessed using a 6 item Likert-type scale developed by Madjar et al. [39], where participants were asked to rate their agreement with statements like “I suggest radically new ways to improve products or services.” The Cronbach’s alpha coefficient yielded a high value of 0.90, reflecting strong internal consistency among the items.

#### 2.2.3. Prosocial Motivation

Prosocial motivation was measured by a five-item scale adapted from Grant and Sumanth [39], which includes items such as “I prefer to work on tasks that allow me to have a positive impact on others”. Cronbach’s alpha coefficient was 0.94.

#### 2.2.4. Prosocial Impact

Prosocial impact was assessed with the 3 item scale developed by Grant [40]. A sample item was “I am aware of how my work today will help others (e.g., colleagues, patients and their family)”. Cronbach’s alpha coefficient was 0.95.

#### 2.2.5. Needs Satisfaction

Needs satisfaction was measured by a scale developed by Huyghebaert-Zouaghi et al. [41] that allows simultaneous assessment of not only need satisfaction and frustration but also need unfulfillment. Cronbach’s alpha coefficients are as follows: Autonomy (3 items; α = 0.89), Competence (3 items; α = 0.87), Relatedness (3 items; α = 0.93). We employed a Likert-type scale ranging from 1 (strong disagreement) to 7 (strong agreement) to measure participants’ responses to the items on this scale.

#### 2.2.6. Benevolence Satisfaction

An adapted version of Martela and Ryan’s [36] scale was used to measure benevolence satisfaction in the workplace. The scale consists of four statements, assessing perceptions of positive impact on others, contribution to society, positive influence on colleagues and clients, and improvement of their well-being. The scale demonstrated satisfactory internal consistency (Cronbach’s α = 0.87). We employed a Likert-type scale ranging from 1 (strong disagreement) to 7 (strong agreement) to measure participants’ responses to the items on this scale.

### 2.3. Data Analysis

The moderated mediation model corresponds to Model 14 in Hayes [42] and is employed to explore the relationships among prosocial impact, need satisfaction, benevolence satisfaction, and work outcomes (innovative work behavior and creativity). The model investigates the indirect effects of prosocial impact on work outcomes, mediated by need satisfaction and benevolence satisfaction. Additionally, the moderating role of prosocial motivation in the relationship between benevolence satisfaction and work outcomes is examined. The indirect effects are calculated using the product of two paths approach [43]. Furthermore, the index of moderated mediation was computed to explore whether the indirect effect of social impact on work outcomes through benevolence satisfaction varies based on levels of prosocial motivation. All variables were standardized to facilitate the interpretation of coefficients and simple (conditional) effects.

For the analyses, we employed the statistical language R v4.3.0 [44] and the structural equation modeling library lavaan v0.16-15 [45]. These robust statistical tools enabled us to conduct thorough examinations of the relationships and interactions between the variables, ensuring rigorous and comprehensive results.

## 3. Results

The final analyses included N = 528 employees, of which 64% were female, from various industries and occupations in Greece (N = 217) and Canada (N = 309). Most of the participants (52.8%) held a master’s degree, with an average age of 37.5 years and an average of 7.3 years of employment.

Before conducting the main analyses, data were examined for confirmatory factorial analysis (CFA), univariate and multivariate normality, and missing values. Kurtosis values were examined for individual variables, and none exceeded the critical threshold of 3.00, indicating that the variables were not severely non-normally distributed [46]. Hence, multivariate normality was not a significant concern in this study. Little’s MCAR test [47] was also performed to see if missing values were completely missing at random, and the test was not significant. Moreover, to investigate potential distinctions between the two populations comprising our sample (Greek and Canadian), we performed an ANOVA test on the variables within our conceptual framework. The results indicated significant variations in participants’ innovative work behaviors (F(1, 519) = 34.98, *p* = 0.001) and creativity (F(1, 519) = 10.22, *p* = 0.001) based on their respective countries.

The results of our preliminary analyses hold relevance for the robustness of our subsequent main analyses. We confirmed that the variables under examination did not exhibit severe non-normality, mitigating concerns regarding multivariate normality. Moreover, the absence of significance in Little’s MCAR test supported the assumption that missing values were not systematically related to the study variables. Additionally, differences observed in innovative work behaviors and creativity based on participants’ country prompted us to include country and other sociodemographic variables as covariates in our subsequent analyses to account for their potential influence.

Table 1 presents the descriptive statistics of the study variables, offering a comprehensive overview of their individual characteristics. Concurrently, Figure 1 portrays a correlogram that depicts the interrelationships between the variables, showcasing the pairwise Pearson correlations among them. Notably, all variables exhibit positive associations, signifying significant statistical differences from zero at the 0.001 level of significance. However, the correlation between autonomy satisfaction and prosocial motivation, though still positively related, demonstrates significance at the 0.01 level (r = 0.11, *p* = 0.009; refer to Figure 1). This finding accentuates the nuanced nature of their association, warranting further exploration and interpretation in light of the study’s objectives.

### 3.1. Effect of Social Impact on Need Satisfaction

The findings of the moderated mediation model are graphically represented in Figure 2. The left-hand side of the figure illustrates that social impact exerts a positive influence on all dimensions of need satisfaction, including autonomy, competence, relatedness, and benevolence.

### 3.2. Effect of Need Satisfaction on Work Outcomes

In terms of the impact of need satisfaction on work outcomes, the right-hand side of Figure 2 reveals that competence satisfaction has a positive effect on both innovative work behavior and creativity. Additionally, autonomy satisfaction positively influences creativity, but not innovative work behavior. Conversely, benevolence satisfaction positively impacts innovative work behavior, while creativity remains unaffected by this need. Notably, when accounting for the effects of the other needs, relatedness satisfaction does not appear to significantly influence any of the work outcomes examined within this study.

### 3.3. The Moderating Role of Prosocial Motivation in the Relationship between Benevolence Satisfaction and Work Outcomes

Moving on to the moderation of benevolence satisfaction on work outcomes by prosocial motivation, it is important to note that prosocial motivation shows a positive association with both work outcomes (innovative work behavior and creativity). However, the moderating role of prosocial motivation in the effects of need satisfaction on work outcomes is limited, as evidenced by non-significant interaction terms for most cases (all *p*-values for interaction terms >0.186). However, a notable exception is observed in the case of the effect of benevolence satisfaction on innovative work behavior. The relationship between benevolence satisfaction and innovative work behavior becomes stronger (more positively pronounced) as prosocial motivation increases (Figure 3). Specifically, as depicted in Figure 2, when prosocial motivation is 1 standard deviation below the mean, the effect of benevolence satisfaction on innovative work behavior is moderate (β = 0.131, SE = 0.054, z = 2.45, *p* = 0.014). At average levels of prosocial motivation, this effect becomes more pronounced (β = 0.213, SE = 0.049, z = 4.35, *p* < 0.001). Moreover, when prosocial motivation is 1 standard deviation above the mean, the effect is even stronger (β = 0.295, SE = 0.054, z = 5.52, *p* < 0.001).

### 3.4. (Moderated) Mediation of the Effect of Prosocial Impact on Work Outcomes through Need Satisfaction

Table 2 presents the standardized indirect effects of social impact on work outcomes, specifically innovative work behavior and creativity, through the mediating influence of need satisfaction. The upper section reveals that prosocial impact exerts a positive impact on innovative work behavior indirectly by positively influencing both competence and benevolence satisfaction. In essence, higher levels of social impact are associated with elevated innovative work behavior, attributed to the simultaneous elevation of competence and benevolence satisfaction. Notably, these indirect effects are consistently observed across all levels of prosocial motivation.

While the indirect effect of social impact on innovative work behavior through competence satisfaction remains unaffected by prosocial motivation, as indicated by a nonsignificant moderated mediation index, the same cannot be said for the indirect effect through benevolence satisfaction (moderated mediation index: 0.059 [SE = 0.016], z = 3.79, *p* < 0.001). Specifically, the indirect effect of social impact on innovative work behavior through benevolence satisfaction strengthens as prosocial motivation increases. This implies that individuals with higher prosocial motivation exhibit even more substantial improvements in innovative work behavior when social impact and benevolence satisfaction jointly contribute to their work experiences. Importantly, after accounting for the effects of need satisfaction on innovative work behavior, social impact no longer exerts a direct effect on the outcome, indicating complete mediation [48].

The lower section of Table 2 demonstrates that social impact also positively affects creativity indirectly by fostering both competence and autonomy satisfaction. Once again, higher levels of social impact are associated with increased competence and autonomy satisfaction, leading to enhanced creativity. Remarkably, these indirect effects remain consistent across all levels of prosocial motivation, as evidenced by the nonsignificant moderated mediation indexes. Furthermore, after controlling for the influence of need satisfaction on creativity, the direct effect of social impact on the outcome is rendered insignificant, confirming complete mediation.

In summary, the findings underscore the significance of need satisfaction as a mediating mechanism through which social impact influences work outcomes. The interplay of competence, benevolence, and autonomy satisfaction plays a crucial role in facilitating innovative work behavior and creativity, providing valuable insights for understanding the role of prosocial motivation in this context.

## 4. Discussion

### 4.1. Theoretical Implications

This study holds theoretical significance for researchers and scholars in the fields of organizational psychology, motivation theory, and workplace behavior.

First, our study is aligned with the foundational tenets of SDT, emphasizing the pivotal role of psychological needs satisfaction in nurturing positive work-related behaviors. Notably, our research extends this paradigm by being the first to incorporate innovative work behavior and creativity as integral components of positive work outcomes within the context of SDT. This expansion enriches our understanding of the mechanisms by which psychological needs satisfaction fosters desirable work-related behaviors, further advancing the theoretical underpinnings of SDT.

Moreover, our research advances the existing body of literature on self-determination theory by using benevolence satisfaction as a mediating variable in the relationship between prosocial impact and work outcomes. Our findings go beyond previous work on benevolence satisfaction [34] by demonstrating its unique indirect influence on the link between prosocial impact and innovative work behaviors. These findings emphasize the importance of considering benevolence satisfaction alongside the three initial psychological needs when exploring their collective impact on work-related outcomes.

Additionally, our demographic analysis provides valuable insights into the nuanced contextual factors influencing the observed relationships. The notable disparities in innovative work behavior across participants from two different countries, as evidenced by our results, underscore the critical importance of considering cultural and contextual dynamics in the landscape of organizational behavior research. This deeper level of understanding ultimately contributes to the refinement and effectiveness of interventions and approaches tailored to the specific needs and expectations of diverse cultural and contextual backgrounds.

In sum, our research contributes to the theoretical advancement of self-determination theory, prosocial motivation, and IWB. The demographic analysis highlights the importance of considering contextual factors in organizational behavior research, offering a bridge between theory and practice. This holistic approach enhances our understanding of the intricate dynamics of motivation and its impact on work outcomes within diverse organizational contexts.

### 4.2. Limitations and Future Research

While this study has provided valuable insights, it is crucial to recognize its limitations and outline potential avenues for future research to address them.

First, the cross-sectional design of our data analysis limits our ability to draw causal conclusions between variables. For instance, it is possible that work outcomes (e.g., IWB) influence psychological states (need satisfaction and benevolence). However, the present study builds on a sequence supported by prior empirical evidence in the self-determination literature [30]: psychological needs satisfaction (or well-being optimizers) → work motivation quality → employee behaviors. Longitudinal research using at least four time points has validated this sequence [49]. To address this limitation, future research could employ longitudinal or experimental data analyses, incorporating multiple time points to validate and strengthen the proposed sequence.

Secondly, our reliance on self-reported data raises the potential concern of common method bias influencing our results. However, it is worth noting that for variables primarily related to psychological experiences and states, such as needs satisfaction, alternative measurement methods may prove challenging and possibly less accurate [50]. Although we conducted a Harman’s one-factor test to assess the magnitude of this bias, revealing an acceptable extracted total variance, future studies could enhance robustness by incorporating objective measures, particularly for variables such as prosocial behaviors, which might benefit from methods like peer observation.

Thirdly, it is crucial to recognize that our study sample consisted of French-speaking Canadians and Greeks. This homogeneity limits the generalizability of our conclusions, primarily applying to these specific cultural groups. Indeed, this diversity becomes evident when we observe that participants with different nationalities, such as Greeks and Canadians, displayed substantial variations in their levels of innovative work behavior (IWB). A growing body of research has indicated that cultural intelligence can mitigate the impact of cultural differences on innovative work behaviors [51,52]. Therefore, future investigations should consider the inclusion of cultural intelligence as a potential moderator in the relationships between social impact, need satisfaction, and IWB.

Fourthly, our findings have limited generalizability across diverse job types and industries. It’s important to note that a substantial number of our participants (52.8%) held master’s degrees, which may not accurately represent professions where advanced education levels are typically lower, such as technical or specialized roles. Additionally, our research emphasizes the fundamental importance of fostering innovative work behavior and highlights the moderating role of prosocial motivation in the relationship between benevolence satisfaction and IWB. However, it’s essential to recognize that individuals have varying levels of prosocial motivation from the start. Some people may naturally be more inclined towards prosocial behaviors, while others may have different motivations. Therefore, when applying our results to different occupational contexts, we should exercise caution, considering these individual variations in prosocial motivation. Future research can enhance our understanding by validating our results on more diverse samples that closely resemble the multifaceted reality of workplaces or focus on specific occupations [53]. This approach will ensure that our findings are more representative and applicable to a broader range of occupational contexts.

Lastly, our study did not directly investigate the potential costs associated with fostering prosocial behavior within an organization, which is crucial from a management perspective. While promoting prosocial actions can yield numerous benefits, it’s essential to strike a balance that aligns with profit-maximizing goals. Future research could delve into analyzing the conditions under which prosocial behavior aligns with profit-maximizing behavior for firms. Such an examination would provide valuable insights into the optimal integration of prosocial initiatives within the business framework, ensuring they contribute positively to both social impact and the organization’s bottom line.

### 4.3. Practical Implications and Conclusion

The practical implications of these findings provide valuable guidance for organizations striving to establish a work environment that cultivates employees’ need satisfaction, sparks creativity, and encourages innovative work behavior.

Organizations can take proactive steps to nurture a sense of positive impact among their employees. Encouraging and supporting initiatives that allow employees to contribute positively to their colleagues, the organization, and the broader community can be highly effective. For instance, establishing programs that facilitate volunteering, mentoring, or social responsibility projects can provide employees with opportunities to make a meaningful difference. This, in turn, enhances their sense of prosocial impact and overall need satisfaction.

To bolster competence and autonomy satisfaction, organizations should prioritize professional development. Providing access to training programs, workshops, and resources enables employees to hone their skills and expand their knowledge base. Simultaneously, granting employees greater decision-making authority over their work processes fosters autonomy and empowers them to take ownership of their tasks. This empowerment contributes significantly to boosting creative contributions.

In addition to focusing on individual growth, organizations can emphasize the value of collaboration, teamwork, and mutual support. Encouraging employees to actively assist and support their colleagues not only builds a sense of benevolence but also drives engagement in innovative initiatives. A workplace culture that prioritizes cooperation and collective problem-solving effectively harnesses employees’ benevolence motivations, leading to enhanced innovative work behavior.

Recognizing the role of prosocial motivation as a moderator, organizations can refine their support and recognition strategies. Recognizing and encouraging employees who consistently demonstrate prosocial behaviors can further boost their engagement in innovative work behavior. Acknowledging and reinforcing prosocial actions aligns with a workplace culture that values contributions to the collective success of the organization.

In conclusion, this study provides insight into the intricate dynamics of prosocial impact, need satisfaction, benevolence satisfaction, prosocial motivation, and their effects on work outcomes. By acknowledging the significance of fostering prosocial impact and creating a work environment that fulfills employees’ psychological needs, organizations can tap into their workforce’s potential to drive innovation and creativity. This strategic approach ultimately contributes to the long-term success and growth of the organization. The practical examples provided offer a more detailed perspective on how these principles can be implemented in real-world organizational settings.

## Figures and Tables

**Figure 1 behavsci-13-00984-f001:**
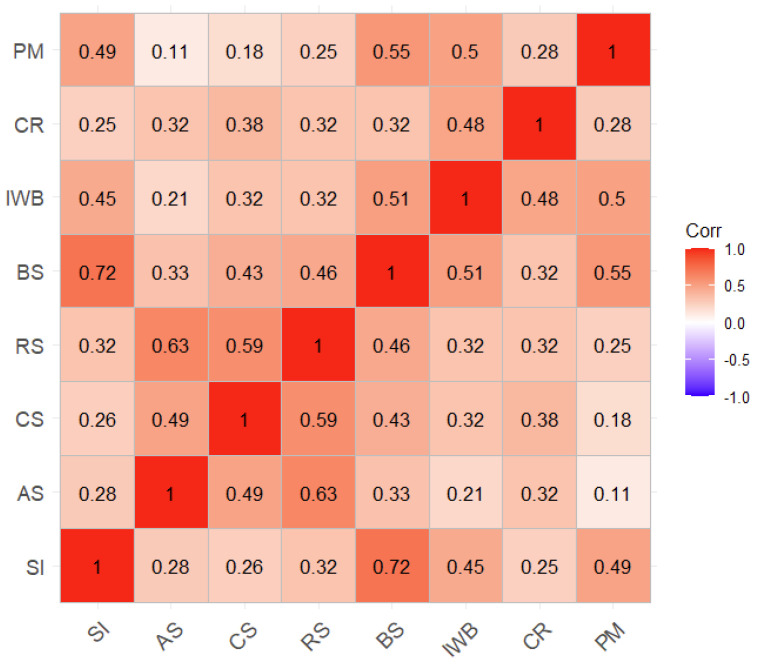
Correlogram (Pearson correlations) of the study variables. Note: all correlations are significantly different from 0. SI = social impact; AS = autonomy satisfaction; CS = competence satisfaction; RS = relatedness satisfaction; BS = benevolence satisfaction; IWB = innovative work behavior; CR = creativity; PM = prosocial motivation.

**Figure 2 behavsci-13-00984-f002:**
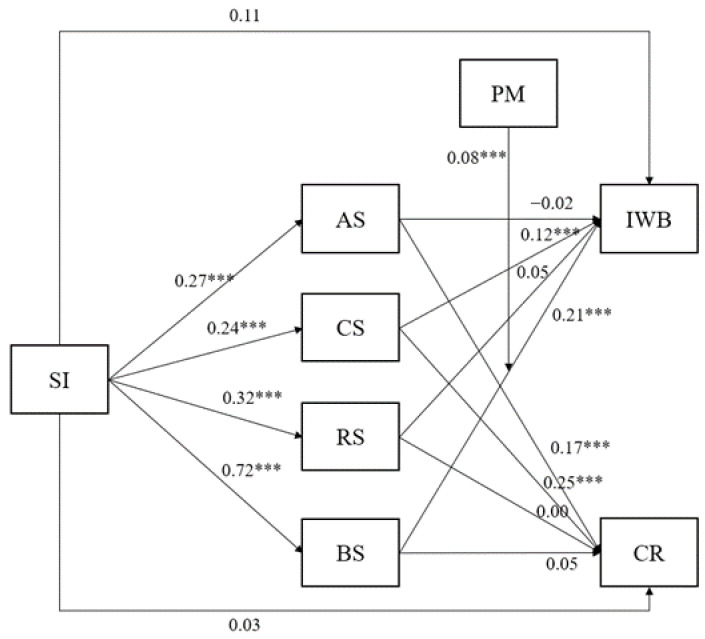
Mediation of the relationship between social impact and work outcomes through need satisfaction, moderated by prosocial motivation. Note. All coefficients are standardized. The non-significant interaction terms involving PM are not shown (only the interaction between PM and BS is shown). The direct effects from PM to the work outcomes are not shown (see text). SI = social impact; AS = autonomy satisfaction; CS = competence satisfaction; RS = relatedness satisfaction; BS = benevolence satisfaction; IWB = innovative work behavior; CR = creativity; PM = prosocial motivation. *** *p* < 0.001.

**Figure 3 behavsci-13-00984-f003:**
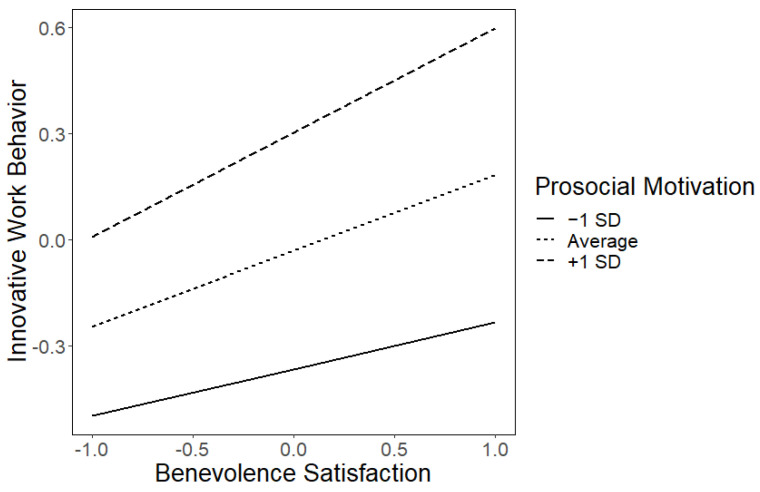
Effect of benevolence satisfaction on innovative work behavior as a function of prosocial motivation. Note. All variables are standardized (z scores). Values are given for average values of the other needs’ satisfaction (autonomy, competence, and relatedness).

**Table 1 behavsci-13-00984-t001:** Descriptive statistics for all study variables.

Variable	N	Min	Max	Mean	SD
Social Impact	549	1.00	7.00	4.96	1.35
Need Satisfaction					
Autonomy	557	1.00	7.00	5.25	1.26
Competence	557	1.00	7.00	5.71	0.98
Relatedness	557	1.33	7.00	5.23	1.14
Benevolence	556	1.00	7.00	5.25	1.07
Work Outcomes					
Innovative Work Behavior	526	1.00	6.89	3.72	1.25
Creativity	527	1.00	7.00	4.92	1.11
Prosocial Motivation	548	1.00	7.00	5.92	1.03

**Table 2 behavsci-13-00984-t002:** Standardized indirect effects and moderated mediation indexes for the mediation of the relationship between social impact and work outcomes via need satisfaction, moderated by prosocial motivation.

					95% Confidence Interval				
Work Outcome	Need	Prosocial Motivation	Standardized Indirect Effect	*p*	Lower Limit	Upper Limit	Mediation?	Moderated Mediation Index	*p*
IWB	Autonomy	Low	0.01	0.583	−0.02	0.04	No	−0.01	0.195
		Average	−0.01	0.559	−0.02	0.01	No		
		High	−0.02	0.183	−0.05	0.01	No		
	Competence	Low	0.03	0.022	0.00	0.06	Yes	0.00	0.935
		Average	0.03	0.003	0.01	0.05	Yes		
		High	0.03	0.028	0.00	0.05	Yes		
	Relatedness	Low	0.03	0.070	0.00	0.06	No	−0.01	0.202
		Average	0.02	0.186	−0.01	0.04	No		
		High	0.00	0.860	−0.03	0.03	No		
	Benevolence	Low	0.09	0.015	0.02	0.17	Yes	0.06	0.000
		Average	0.15	0.000	0.08	0.22	Yes		
		High	0.21	0.000	0.14	0.29	Yes		
CR	Autonomy	Low	0.04	0.010	0.01	0.08	Yes	0.00	0.909
		Average	0.05	0.000	0.02	0.07	Yes		
		High	0.05	0.007	0.01	0.08	Yes		
	Competence	Low	0.06	0.001	0.02	0.09	Yes	0.01	0.575
		Average	0.06	0.000	0.03	0.09	Yes		
		High	0.07	0.000	0.03	0.10	Yes		
	Relatedness	Low	0.01	0.718	−0.03	0.04	No	−0.01	0.495
		Average	0.00	0.911	−0.03	0.02	No		
		High	−0.01	0.596	−0.04	0.02	No		
	Benevolence	Low	0.04	0.375	−0.05	0.12	No	0.00	0.771
		Average	0.03	0.401	−0.04	0.11	No		
		High	0.03	0.513	−0.06	0.11	No		

Note: low and high levels of prosocial motivation correspond to 1 standard deviation below and above the mean, respectively. Significant effects are in bold. IWB = innovative work behavior; CR = creativity.

## Data Availability

Data are contained within the article.

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
