# Peer review of "Positive Impact, Creativity, and Innovative Behavior at Work: The Mediating Role of Basic Needs Satisfaction"

_behavsci, 2023, doi:10.3390/bs13120984_

Round 1
Reviewer 1 Report
Comments and Suggestions for Authors
This manuscript investigates the impacts of social impact on employees’ innovative work behavior and creativity, and tests the mediating role of needs satisfaction. I have some concerns regarding this manuscript and have summarized them below.
1. The alignment between the manuscript's title and its main content appears inadequate. It may be advisable for the authors to contemplate a more precise title.
2. Creativity involves the generation of novel ideas by individuals or teams, while innovative work behavior encompasses the subsequent actions required to introduce or implement these novel ideas (Woodman R W, Sawyer J E, Griffin, 1993). Therefore, creativity is commonly seen as the initial step toward engaging in innovative work behavior (Amabile, 1996). Some scholars have even directly considered creativity a significant manifestation of innovative behavior (Shin S J, Yuan F, Zhou, 2017). Consequently, using creativity and innovative work behavior simultaneously as outcomes is not advisable. The authors are advised to select one of these two outcomes.
3. Could the authors please clarify the theoretical contributions of this manuscript? The authors have not adequately highlighted them.
References:
Amabile, T. M. 1996. Creativity in context. Boulder, CO: Westview.
Shin, S. J., Yuan, F., & Zhou, J. (2017). When perceived innovation job requirement increases employee innovative behavior: A sensemaking perspective. Journal of Organizational behavior, 38(1), 68-86.
Woodman, R. W., Sawyer, J. E., & Griffin, R. W. (1993). Toward a theory of organizational creativity. Academy of management review, 18(2), 293-321.
Author Response
October 14th, 2023
Dear evaluation committee,
With this letter, we resubmit our article “Positive Impact, Creativity and Innovative Behavior at Work: The mediating role of basic needs satisfaction”, which has been revised based on the feedback from all reviewers. In this letter, we provide our responses along with our explanations for modifications made in line with each comment. We would like to thank you for pointing out some important issues that have contributed our article’s improvement.
We hope the changes made to the article meet your expectations and we welcome your future comments.
The authors
Based on the comments and the evaluation of all three reviewers we modified the manuscript in the following ways:
The article that has now an updated title “Positive Impact, Creativity and Innovative Behavior at Work: The mediating role of basic needs satisfaction”, has been succinctly contextualized with respect to previous theoretical background and empirical research on the topic. More precisely, two paragraphs (see lines 43-58) highlighted the differentiated nature of IWB and creativity while in parallel the notion of prosocial impact and motivation has been linked with the work outcomes under a more detailed explanation on the process of fostering creativity through experiencing prosocial motivation and perceived impact has been described (see lines 71-89). Over the manuscript, 15 additional references have been added so as to enrich the theoretical arguments both in introduction and discussion part (see lines 548-578)[1]. The section “Results” have been adapted based on all reviewers’ comments (i.e., demographics, description of scales, CFA).
We have also incorporated the discussion of theoretical implications into the newly added section titled "Theoretical Implications " (see lines 360-391). This section comprehensively addresses how our research findings can benefit both theorists and practitioners, ensuring that it covers the insights gained from the demographic analysis of our sample. We have reworked the "Practical Implications and Conclusion" section to include more practical examples, as requested (see lines 419-443). This revised section provides concrete illustrations of how organizations can apply the research findings in real-world contexts, offering valuable insights for managers and practitioners seeking actionable guidance. Lastly, all minor changes (i.e., some clarifications about scales, some proposals about the text flow and 3 typos) have been taken into consideration and the manuscript has been correspondingly updated.
Reviewer 1
This manuscript investigates the impacts of social impact on employees’ innovative work behavior and creativity, and tests the mediating role of needs satisfaction. I have some concerns regarding this manuscript and have summarized them below.
Thanks a lot for your interest and your insightful comments.
The alignment between the manuscript's title and its main content appears inadequate. It may be advisable for the authors to contemplate a more precise title.
Thanks for pointing this crucial issue. The new title is now representing in a more coherent way the concrete ideas and research in the manuscript.
- Creativity involves the generation of novel ideas by individuals or teams, while innovative work behavior encompasses the subsequent actions required to introduce or implement these novel ideas (Woodman R W, Sawyer J E, Griffin, 1993). Therefore, creativity is commonly seen as the initial step toward engaging in innovative work behavior (Amabile, 1996). Some scholars have even directly considered creativity a significant manifestation of innovative behavior (Shin S J, Yuan F, Zhou, 2017). Consequently, using creativity and innovative work behavior simultaneously as outcomes is not advisable. The authors are advised to select one of these two outcomes.
We have provided some arguments for this decision by adding some arguments in the introductory text (see lines 43-58). Moreover, according to recent metanalyses (see Farrukh et al., 2023) the inclusion of idea development and implementation in IWB broadens the concept beyond creativity.
Moreover, in terms of the analysis we can support this decision by the following facts:
- We have evidence that the two measures assess two different constructs in our study. Indeed, the correlation between the two outcomes is .48, which is much lower than what is usually deemed representative of a single construct (.95 or .90), and the correlation does not stand out from the other correlations in the current study (Figure 1).
- As both outcomes only play the role of outcome in the model without ever predicting other variables (i.e., they are purely endogeneous), having both outcomes in the same model does not affect the results in the other parts of the model. To be sure, we've verified this, and results are exactly identical when we remove either one of the outcomes (in terms of coefficients, hypothesis tests, p-values, etc.).Considering that we have no indication that they represent the same construct, and that including both does not affect results in any way, we prefer providing both measures to interested readers and future researchers.
- Could the authors please clarify the theoretical contributions of this manuscript? The authors have not adequately highlighted them.
Thanks a lot for raising this issue. We have also incorporated the discussion of theoretical implications into the newly added section titled "Theoretical Implications " (see lines 360-391). This section comprehensively addresses how our research findings can benefit both theorists and practitioners, ensuring that it covers the insights gained from the demographic analysis of our sample.
References:
Farrukh, M., Meng, F., Raza, A., & Wu, Y. (2023). Innovative work behaviour: The what, where, who, how and when. Personnel Review, 52(1), 74–98. https://doi.org/10.1108/PR-11-2020-0854
[1] We have not formulated the references based on the journal template/style so as to adapt all text during the final version and facilitate reviewers to spot more easy the additions.

Reviewer 2 Report
Comments and Suggestions for Authors
Author Response
October 14th, 2023
Dear evaluation committee,
With this letter, we resubmit our article “Positive Impact, Creativity and Innovative Behavior at Work: The mediating role of basic needs satisfaction”, which has been revised based on the feedback from all reviewers. In this letter, we provide our responses along with our explanations for modifications made in line with each comment. We would like to thank you for pointing out some important issues that have contributed our article’s improvement.
We hope the changes made to the article meet your expectations and we welcome your future comments.
The authors
Based on the comments and the evaluation of all three reviewers we modified the manuscript in the following ways:
The article that has now an updated title “Positive Impact, Creativity and Innovative Behavior at Work: The mediating role of basic needs satisfaction”, has been succinctly contextualized with respect to previous theoretical background and empirical research on the topic. More precisely, two paragraphs (see lines 43-58) highlighted the differentiated nature of IWB and creativity while in parallel the notion of prosocial impact and motivation has been linked with the work outcomes under a more detailed explanation on the process of fostering creativity through experiencing prosocial motivation and perceived impact has been described (see lines 71-89). Over the manuscript, 15 additional references have been added so as to enrich the theoretical arguments both in introduction and discussion part (see lines 548-578)[1]. The section “Results” have been adapted based on all reviewers’ comments (i.e., demographics, description of scales, CFA).
We have also incorporated the discussion of theoretical implications into the newly added section titled "Theoretical Implications " (see lines 360-391). This section comprehensively addresses how our research findings can benefit both theorists and practitioners, ensuring that it covers the insights gained from the demographic analysis of our sample. We have reworked the "Practical Implications and Conclusion" section to include more practical examples, as requested (see lines 419-443). This revised section provides concrete illustrations of how organizations can apply the research findings in real-world contexts, offering valuable insights for managers and practitioners seeking actionable guidance. Lastly, all minor changes (i.e., some clarifications about scales, some proposals about the text flow and 3 typos) have been taken into consideration and the manuscript has been correspondingly updated.
-------------------------------------------------------------------------------------------------------
Reviewer 2
This paper deals with the quite important and essential issue in analyzing the creativity. In this sense, I evaluate this paper highly for publishing in the journal. Especially, the hypothesis that social experience and prosocial behavior links to creativity has profound implications for the companies to promote creativities.
Your comments are much appreciated and we agree that this stream of research can substantially foster the discussion about how other non-tangible “rewards” can foster IWB and creativity.
l have several comments:
- We need more detailed explanation on the process of fostering creativity through experiencing pro-social activities. In the paper, it is described on this as follows: "Prosocial motivation refers to the desire to act for the benefit or welfare of others and has been linked to a wide array of positive personal and professional outcomes since it helps employees go beyond the limitations of their own perspectives, improve their sensitivity to the needs of others, and perform tasks to the best of their abilities and interests all of which are crucial for promoting innovativeness and creativity."The point related with "fostering creativity" is "improve their sensitivity to the needs of others". If there are some explanations on this points in the psychological literature, we would be happy to know them.
Thanks for this very important suggestion. As you can notice in the introductory part, we have added two paragraphs based on your comments (see lines 71-89) and we hope this clarifies in a coherent way the process of fostering creativity through experiencing pro-social activities.
We need to notice, that based on widely used definition of Janssen (Janssen, 2000, p. 288), IWB refers to “the intentional creation, introduction, and application of new ideas within a work role, group or organization, in order to benefit role performance, the group, or the organization”. Built into this definition are two basic elements: creativity/innovation and the idea of benefiting others within the organization. From a motivational standpoint, the question is whether these two distinct elements have a common motivational basis. One type of approach, offered by social exchange theories, is that, through IWB, employees reciprocate benefits received from the organization. From this perspective, there are environmentally based motivators, such as rewards, that drive IWB (e.g., Atitumpong & Badir, 2018). Another type of approach focuses on autonomous forms of motivation, such as intrinsic motivation (e.g., Devloo et al., 2015) and treats IWB as primarily motivated by the self, instead of external motivators. While both perspectives have their merits, our focus in this research is on the prosocial motivation of IWB.
- Atitumpong, A., & Badir, Y. F. (2018). Leader-member exchange, learning orientation and innovative work behavior. Journal of Workplace Learning, 30(1), 32–47. https://doi.org/10.1108/JWL-01-2017-0005
- Devloo, T., Anseel, F., De Beuckelaer, A., & Salanova, M. (2015). Keep the fire burning: Reciprocal gains of basic need satisfaction, intrinsic motivation and innovative work behaviour. European Journal of Work and Organizational Psychology, 24(4), 491–504. https://doi.org/10.1080/1359432X.2014.931326
- Janssen, O. (2000). Job demands, perceptions of effort-reward fairness and innovative work behaviour. Journal of Occupational and Organizational Psychology, 73(3), 287–302. https://doi.org/10.1348/096317900167038
- It would be necessary to describe the impact of the high percentage of master's degree holders (52.8%). It is necessary to discuss in what capacity and in what types of jobs this hypothesis is valid. In addition, the hypothesis may depend on the job type or occupations. For example, it is not obvious that the similar hypothesis would hold true for engineers or research workers. lt is not also obvious that "Exploration, generation, championing, and implementation of ideas" are applicable regardless of job type.
This comment is much appreciated and this is why has been explicitly highlighted in the limitations section. This study’ s focus was on motivational aspects of IWB and it would be very challenging to include occupations as a control variable since the cultural and occupational context in Canada and Greece substantially differs and many other details also intervenes (e.g. public/private sector). While other demographics (i.e., gender, educational level, tenure) were easily included in the analyses, occupations pose several limitations. This is why we have spotted some strategies that future research can employ.
The additional text on the limitation part follows:
Fourthly, our findings have limited generalizability across diverse job types and industries. It's important to note that a substantial number of our participants (52.8%) held master's degrees, which may not accurately represent professions where advanced education levels are typically lower, such as technical or specialized roles. Additionally, our research emphasizes the fundamental importance of fostering innovative work behavior and highlights the moderating role of prosocial motivation in the relationship between benevolence satisfaction and IWB. However, it's essential to recognize that individuals have varying levels of prosocial motivation from the start. Some people may naturally be more inclined towards prosocial behaviors, while others may have different motivations. Therefore, when applying our results to different occupational contexts, we should exercise caution, considering these individual variations in prosocial motivation. Future research can enhance our understanding by validating our results on more diverse samples that closely resemble the multifaceted reality of workplaces or focus on specific occupations (e.g. Gkontelos et el., 2023). This approach will ensure that our findings are more representative and applicable to a broader range of occupational contexts.
Gkontelos, A.; Vaiopoulou, J.; Stamovlasis, D. Teachers’ Innovative Work Behavior as a Function of Self-Efficacy, Burnout, and Irrational Beliefs: A Structural Equation Model. Eur. J. Investig. Health Psychol. Educ. 2023, 13, 403–418. https://doi.org/ 10.3390/ejihpe13020030
- From the management perspective, we need to examine the possibility of increasing costs from behaving pro-socially. It would be necessary to analyse the condition for the pro social behavior to be consistent with the profit maximizing behavior at the
Thank you for raising this important point. We have addressed the issue of potential costs associated with prosocial behavior in the "Limitations and Future Research" section of our discussion. Your insight is greatly appreciated.
Thanks once again for your insightful comments and valuable suggestions.
[1] We have not formulated the references based on the journal template/style so as to adapt all text during the final version and facilitate reviewers to spot more easy the additions.

Reviewer 3 Report
Comments and Suggestions for Authors
Dear Authors,
I have reviewed your work with great interest, and I appreciate the theoretical and practical contributions it offers. However, I have some concerns and suggestions that I believe will enhance the clarity and impact of your article.
1. Sample Selection and Data Collection:
It is essential to provide a comprehensive explanation of your sample selection process, especially since you collected data from two different countries. Please clarify the rationale behind selecting these countries and how their differences in employability, salaries, and working conditions were accounted for in your study.
Specify the industries from which you collected data and provide the exact number of participants from Greece and Canada.
2. Measures:
It appears that you used various Likert-type scales (3, 5, 6, and 7 points) in your study. Please remove the text under the "Measures" subsection and specify the exact scales used for assessing "needs satisfaction" and "benevolence satisfaction."
3. Data Analysis:
The demographic results of your sample should be moved to the Results section, accompanied by a discussion of their relevance and potential use as covariates. Additionally, incorporate research questions related to these results into the Introduction section.
4. Table 1:
In Table 1, make it explicit which Likert scales were used to assess each concept under investigation (e.g., Innovative Work Behavior on a 7-point scale).
Procosial motivation was measured through a 5-point scale (lines 171-174). This result refer to a 7-point scale –this is the case in other measures as well. If so, then the above comment of mine (about the delation of the text referring to the 7 point scale) should remain as it is, and corrections should be made to each of the measures (e.g. Creativity, Prosocial motivation, and so on)
5. Discussion:
Incorporate practical examples in the discussion to illustrate how organizations can apply your research findings. Provide concrete examples of practical implementations that can be utilized by managers and practitioners.
Discuss theoretical applications and how your results can benefit both theorists and practitioners.
Ensure that the discussion includes the results related to the demographic analysis of the sample.
6. Limitations:
Address the limitations of your study, including the potential limitations arising from collecting data from two different working cultures and the relatively small sample size, considering the diverse origins of the participants.
General comments
Line 82: the three fundamental psychological needs are as follows. “The” instead of “the”
Line 110: add the pages of the reference put into quotation marks
Line 127. Correct: promotimig innovativeness and
Incorporating these changes will enhance the overall quality and impact of your work. Thank you for your valuable contribution to the field.
Author Response
October 14th, 2023
Dear evaluation committee,
With this letter, we resubmit our article “Positive Impact, Creativity and Innovative Behavior at Work: The mediating role of basic needs satisfaction”, which has been revised based on the feedback from all reviewers. In this letter, we provide our responses along with our explanations for modifications made in line with each comment. We would like to thank you for pointing out some important issues that have contributed our article’s improvement.
We hope the changes made to the article meet your expectations and we welcome your future comments.
The authors
Based on the comments and the evaluation of all three reviewers we modified the manuscript in the following ways:
The article that has now an updated title “Positive Impact, Creativity and Innovative Behavior at Work: The mediating role of basic needs satisfaction”, has been succinctly contextualized with respect to previous theoretical background and empirical research on the topic. More precisely, two paragraphs (see lines 43-58) highlighted the differentiated nature of IWB and creativity while in parallel the notion of prosocial impact and motivation has been linked with the work outcomes under a more detailed explanation on the process of fostering creativity through experiencing prosocial motivation and perceived impact has been described (see lines 71-89). Over the manuscript, 15 additional references have been added so as to enrich the theoretical arguments both in introduction and discussion part (see lines 548-578)[1]. The section “Results” have been adapted based on all reviewers’ comments (i.e., demographics, description of scales, CFA).
We have also incorporated the discussion of theoretical implications into the newly added section titled "Theoretical Implications " (see lines 360-391). This section comprehensively addresses how our research findings can benefit both theorists and practitioners, ensuring that it covers the insights gained from the demographic analysis of our sample. We have reworked the "Practical Implications and Conclusion" section to include more practical examples, as requested (see lines 419-443). This revised section provides concrete illustrations of how organizations can apply the research findings in real-world contexts, offering valuable insights for managers and practitioners seeking actionable guidance. Lastly, all minor changes (i.e., some clarifications about scales, some proposals about the text flow and 3 typos) have been taken into consideration and the manuscript has been correspondingly updated.
Reviewer 3
Dear Authors,
I have reviewed your work with great interest, and I appreciate the theoretical and practical contributions it offers. However, I have some concerns and suggestions that I believe will enhance the clarity and impact of your article.
Thanks for your kind words and encouragement.
Sample Selection and Data Collection:
It is essential to provide a comprehensive explanation of your sample selection process, especially since you collected data from two different countries. Please clarify the rationale behind selecting these countries and how their differences in employability, salaries, and working conditions were accounted for in your study. Specify the industries from which you collected data and provide the exact number of participants from Greece and Canada.
Thanks for pointing this out. In the corresponding section we have provided more details in terms of the demographics (see lines 176 and 237 onwards). In terms of the sample selection, we selected workers from Greece and Canada as they are the home country of the authors, which were financed by a grant for the European union to stimulate international collaboration. Since the goal was to find the universal underlying mechanisms, and also because Self-Determination Theory principles have been tested in several dozen countries, we merged the samples of the two countries together to increase the statistical power. For example, of such research, we initially validated our Multidimensional Work Motivation Scale in 7 different languages and 9 countries (Gagné et al., 2015) and there were negligible or no variabilities in between the different languages/cultures; this scale is now available in 25 different languages.
Gagné, M., Forest, J., Vansteenkiste, M., Crevier-Braud, L., Van den Broeck, A., Aspeli, A. K., Bellerose, J., Benabou, C., Chemolli, E., Güntert, S. T., Halvari, H., Johnson, P., Indiyastuti, D. L., Molstad, M., Naudin, M., Ndao, A., Olafsen, A. H., Roussel, P., Wang, Z., Westbye, C. (2015). The Multidimensional Work Motivation Scale: Validation evidence in seven Languages and nine countries. European Journal of Work and Organizational Psychology, 24, 178-196. http://dx.doi.org/10.1080/1359432X.2013.877892
Measures:It appears that you used various Likert-type scales (3, 5, 6, and 7 points) in your study. Please remove the text under the "Measures" subsection and specify the exact scales used for assessing "needs satisfaction" and "benevolence satisfaction."
Thanks for highlighting this issue. In the results section the measures description is now coherent and easy to understand (see lines 184-220).
Data Analysis:
The demographic results of your sample should be moved to the Results section, accompanied by a discussion of their relevance and potential use as covariates. Additionally, incorporate research questions related to these results into the Introduction section.
Thank you for your suggestions. We have indeed relocated the demographic results to the Results section and provided a detailed discussion of their relevance, particularly in the context of potential use as covariates in our analyses. However, after careful consideration, we have chosen not to incorporate research questions related to these demographic results into the Introduction section. While we appreciate the importance of addressing these aspects comprehensively, introducing research questions based on post-data analysis findings could introduce bias and compromise the scientific rigor of our study. We believe that maintaining the integrity of our initial hypotheses is crucial for maintaining the scientific validity of our research.
Table 1:
In Table 1, make it explicit which Likert scales were used to assess each concept under investigation (e.g., Innovative Work Behavior on a 7-point scale).
Thank you for your feedback. We have updated the "Measures" subsection to specify the exact Likert-type scale used for assessing "needs satisfaction" and "benevolence satisfaction" as requested. Nevertheless we need to notice that the same agreement Likert scale was used to measure creativity, prosocial motivation and impact, and needs satisfaction. They all have 7 points. When we say 5-item scale, we are referring to the 5 items (or statements) that participants answered. They indicated their level of agreement with each item on a 7-point likert scale.
Discussion:
Incorporate practical examples in the discussion to illustrate how organizations can apply your research findings. Provide concrete examples of practical implementations that can be utilized by managers and practitioners.
We have reworked the "Practical Implications and Conclusion" section to include more practical examples, as requested. This revised section provides concrete illustrations of how organizations can apply the research findings in real-world contexts, offering valuable insights for managers and practitioners seeking actionable guidance.
Discuss theoretical applications and how your results can benefit both theorists and practitioners./ Ensure that the discussion includes the results related to the demographic analysis of the sample
We have also incorporated the discussion of theoretical implications into the newly added section titled "Theoretical Implications " (see lines 360-391). This section comprehensively addresses how our research findings can benefit both theorists and practitioners, ensuring that it covers the insights gained from the demographic analysis of our sample.
Limitations:
Address the limitations of your study, including the potential limitations arising from collecting data from two different working cultures and the relatively small sample size, considering the diverse origins of the participants.
Thanks indeed for pointing this out. We have addressed the limitations of our study, including those arising from collecting data from two different working cultures and the relatively small sample size, especially considering the diverse origins of the participants, by expanding the "Limitations and Future Research" section as requested.
General comments
Line 82: the three fundamental psychological needs are as follows. “The” instead of “the”DONE
Line 110: add the pages of the reference put into quotation marks DONE
Line 127. Correct: promotimig innovativeness and, DONE
Incorporating these changes will enhance the overall quality and impact of your work. Thank you for your valuable contribution to the field.
Thanks for your valuable comments and contribution.
The authors
[1] We have not formulated the references based on the journal template/style so as to adapt all text during the final version and facilitate reviewers to spot more easy the additions.

Round 2
Reviewer 1 Report
Comments and Suggestions for Authors
I appreciate the author's response. Nevertheless, these responses still fall short of addressing the concerns within this research paper. Specifically, it is not advisable to use both creativity and innovative work behavior as simultaneous outcomes. Additionally, why did the author solely focus on investigating the conditional boundary of the relationship between benevolence satisfaction and innovative work behavior?
Author Response
October 31th, 2023
Dear reviewer,
Thanks once more for your insightful comments. In this letter, we provide our responses along with our explanations for modifications made in line with each comment. We would like to thank you for pointing out these issues that have already contributed to our article’s improvement.
Your sincerely
The authors
-----------------------------------------------------------------------------------------
Relating your first comment (“Specifically, it is not advisable to use both creativity and innovative work behavior as simultaneous outcomes ») our decision to keep both DVs was based on the following reasons:
Firstly, according to one of the first definitions of IWB by Scott and Bruce (1994), IWB is more than creativity although creativity might be a necessary initial part of IWB, especially in the beginning, in order to generate new and useful ideas. However, IWB is broader than creativity as it also includes the idea promotion and implementation phase (see see Farrukh et al., 2023). In the same vein, research has indicated that organisational practices can have different effects on idea generation and on idea implementation. For example, that task complexity has a negative effect on creativity and idea generation, but a positive effect on idea implementation (Ohly et al., 2006; Urbach et al., 2010). Another study, also, shows that perceptions of training and development have a significant effect on idea generation but not on idea promotion and idea implementation (Veenendaal and Bondarouk, 2015).
Overall, in prior literature, both concepts, creativity (Zhou and George, 2001), and IWB (Scott and Bruce 1994), are separately disclosed which represented a sequence to convert a new idea into an execution method (Van de Ven, 1999, Woodman et al.,1993). For instance, in the first phase: creativity presents to the generation of novel and executable ideas, to introduce a new process, technique, or method to an organization (Amabile and Pratt, 2016). The second phase: IWB demonstrates the execution of creative ideas (Scott & Bruce, 1994), and implementing a solution to a problem at the job or organizational level (Shalley and Gilson, 2004). That is probably why Slåtten and Mehmetoglu (2011) found that creativity alone explained about 43% of the variance of IWB .
Moreover, there is relevant research/approach that supports this decision and has been taken into consideration. We hereby provide some relevant articles that highlighted this conceptual and practical difference between the two variables and have employed a similar approach:
- Oh, SH., Hur, WM. & Kim, H. Employee creativity in socially responsible companies: Moderating effects of intrinsic and prosocial motivation. Curr Psychol42, 18178–18196 (2023).
- Garlatti Costa, G., Bortoluzzi, G.and ÄŒerne, M. (2023), "Can innovative work behaviour spur creativity while working remotely? The role of work–home conflict and social isolation", Management Research Review, Vol. 46 No. 8, pp. 1132-1148.
- Srirahayu DP, Ekowati D, Sridadi AR. Innovative work behavior in public organizations: A systematic literature review. Heliyon. 2023 Feb 8;9(2):e13557. doi: 10.1016/j.heliyon.2023.e13557. PMID: 36852050; PMCID: PMC9958459.
Based on these data we added two paragraphs (see lines 43-58) that highlighted the differentiated nature of IWB and creativity.
In terms of the analysis we can support this decision by the following facts:
- We have evidence that the two measures assess two different constructs in our study. Indeed, the correlation between the two outcomes is .48, which is much lower than what is usually deemed representative of a single construct (.95 or .90), and the correlation does not stand out from the other correlations in the current study (Figure 1).
- As both outcomes only play the role of outcome in the model without ever predicting other variables (i.e., they are purely endogeneous), having both outcomes in the same model does not affect the results in the other parts of the model. To be sure, we've verified this, and results are exactly identical when we remove either one of the outcomes (in terms of coefficients, hypothesis tests, p-values, etc.). Considering that we have no indication that they represent the same construct, and that including both does not affect results in any way, we prefer providing both measures to interested readers and future researchers.
In terms of the second comment, (“why did the author solely focus on investigating the conditional boundary of the relationship between benevolence satisfaction and innovative work behavior ») we would like to thank you for raising this up. We thought that it would not be advisable to explore all needs in the same way since it is now evident that benevolence has a differentiated effect on work outcomes compared to other needs (Martela & Ryan, 2016). More precisely, previous research has demonstrated that benevolence satisfaction—the sense of having a positive impact on other people—can have a unique (when we control for other needs) positive effect on well-being (Martela & Ryan, 2016, 2020) and in this current study we envisage to test and expand these results to some other work outcomes (i.e, creativity and IWB). We focus more on the conditional boundary of the relationship between benevolence satisfaction and IWB because, according to SDT (see Forest et al., 2023) BNS at workplace is an explanatory factor between autonomy supportive behaviors and perceptions (e.g leadership, Messman et al., 2022) and work outcomes.
References:
- Amabile, T. M., & Pratt, M. G. (2016). The dynamic componential model of creativity and innovation in organizations: Making progress, making meaning. Research in organizational behavior, 36, 157-183
- Forest, Jacques, and others, 'Shaping Tomorrow’s Workplace by Integrating Self-Determination Theory: A Literature Review and Recommendations',in Richard M. Ryan (ed.), The Oxford Handbook of Self-Determination Theory (2023; online edn, Oxford Academic.
- Martela, F., & Ryan, R. M. (2020). "Distinguishing between basic psychological needs and basic wellness enhancers: The case of beneficence as a candidate psychological need": Correction. Motivation and Emotion, 44(1), 134. https://doi.org/10.1007/s11031-020-09823-9
- Martela F., Ryan R. M. (2016). The benefits of benevolence: Basic psychological needs, beneficence, and the enhancement of well-being. Journal of Personality, 84(6), 750–764.
- Messmann, G., Evers, A., & Kreijns, K. (2022). The role of basic psychological needssatisfaction in the relationship between transformational leadership and innovative work behavior.HumanResource Development Quarterly,33(1), 29–45
- Farrukh, M., Meng, F., Raza, A., & Wu, Y. (2023). Innovative work behaviour: The what, where, who, how and when. Personnel Review, 52(1), 74–98. https://doi.org/10.1108/PR-11-2020-0854
- Ohly, S., Sonnentag, S. and Pluntke, F. (2006), “Routinization, work characteristics and their relationships with creative and proactive behaviors”, Journal of Organizational Behaviour, Vol. 27 No. 3, pp. 257-279.
- Scott, S.G. and Bruce, R.A. (1994), “Determinants of innovative behavior: a path model of individual innovation in the workplace”, Academy of Management Journal, Vol. 37 No. 3, pp. 580-607.
- Shalley, C. E., Zhou, J., & Oldham, G. R. (2004). The effects of personal and contextual characteristics on creativity: Where should we go from here? Journal of Management, 30, 933-958.
- Slåtten, T., & Mehmetoglu, M. (2011). What are the drivers for innovative behavior in frontline jobs? A study of the hospitality industry in Norway. Journal of Human Resources in Hospitality & Tourism, 10(3), 254-272.
- Slåtten, T., Svensson, G., & Sværi, S. (2011). Empowering leadership and the influence of a humorous work climate on service employees' creativity and innovative behaviour in frontline service jobs. International Journal of Quality and Service Sciences.
- Urbach, T., Fay, D. and Gora, A. (2010), “Extending the job design perspective on individual innovation: exploring the effect of group reflexivity”, Journal of Occupational and Organizational Psychology, Vol. 83 No. 4, pp. 1053-1064.
- Van de Ven, A.H. (1999). The innovation journey. New York: Oxford University Press.
- Veenendaal, A. and Bondarouk, T. (2015), “Perceptions of HRM and their effect on dimensions of innovative work behaviour: evidence from a manufacturing firm”, Management Revue, Vol. 26 No. 2, pp. 138-160.
- Woodman, R. W., Sawyer, J. E., & Griffin, R. W. (1993). Toward a theory of organizational creativity. Academy of Management Review, 18, 293–321.
- Zhou, J., & George, J. M. (2001). When job dissatisfaction leads to creativity: Encouraging the expression of voice. Academy of Management Journal, 44, 682–697